# Memory effects of Eurasian land processes cause enhanced cooling in response to sea ice loss

Tetsu Nakamura [1]*, Koji Yamazaki[1], Tomonori Sato [1] & Jinro Ukita[2]

Amplified Arctic warming and its relevance to mid-latitude cooling in winter have been intensively studied. Observational evidence has shown strong connections between decreasing sea ice and cooling over the Siberian/East Asian regions. However, the robustness of such connections remains a matter of discussion because modeling studies have shown divergent and controversial results. Here, we report a set of general circulation model experiments specifically designed to extract memory effects of land processes that can amplify sea ice–climate impacts. The results show that sea ice–induced cooling anomalies over the Eurasian continent are memorized in the snow amount and soil temperature fields, and they reemerge in the following winters to enhance negative Arctic Oscillation-like anomalies. The contribution from this memory effect is similar in magnitude to the direct effect of sea ice loss. The results emphasize the essential role of land processes in understanding and evaluating the Arctic–mid-latitude climate linkage.

[1] Faculty of Environmental Earth Science, Hokkaido University, Hokkaido 060-0810, Japan. [2] Faculty of Science, Niigata University, Niigata 950-2181, Japan. *email: nakamura.tetsu@ees.hokudai.ac.jp

Acceleration of warming over the Arctic region, a symbol of progressing global warming, has serious local and remote impacts on society and the environment. In the past decade, severe winters have affected densely populated areas in the Northern Hemisphere, such as North America, Europe, and East Asia[1–3]. This contrast between Arctic warming and mid-latitude cooling can be placed in the framework of the negative phase of the Arctic Oscillation (AO)/North Atlantic Oscillation (NAO). Observation-based studies have suggested a strong connection between the rapid decline of Arctic sea ice and wintertime mid-latitude cooling over the Siberian/East Asian regions in association with the negative AO/NAO[4,5]. However, the winter AO/NAO index has shown a positive trend after the strongest negative AO/NAO year of 2010, while Arctic sea ice has remained at a continuously low level. Some modeling studies have provided supporting evidence for this connection and the importance of Arctic sea ice loss[6–8], whereas others have argued that Arctic sea ice loss has only small impacts on mid-latitude climate[9–11]. Therefore, large uncertainty exists among simulation models/studies[12,13], while the underestimation and/or model dependency of the sea ice impacts are suggested as a possible cause[14,15].

Atmosphere–ocean coupling and stratosphere–troposphere coupling processes have been studied extensively in connection with Arctic climate changes. Ice-albedo feedback in the atmosphere–ice–ocean coupling process is essential in Arctic amplification, providing a long-term memory effect from the large heat capacity of the ocean and the sensitivity of the surface

energy balance to albedo[16]. Once sea ice is reduced significantly, stratosphere–troposphere coupling plays a key role in connecting sea ice variations to AO/NAO-like circulation anomalies on intraseasonal timescales[17–19]. A possible role of land processes has also been considered. The relevance of anomalies in autumn snow cover extent over the Eurasian continent to winter negative AO/NAO via a stratospheric pathway is well documented[1,20]. Furthermore, a land process with a longer memory than the atmosphere could affect the atmosphere on a timescale beyond a season[21]. Nevertheless, there are no studies that have examined the memory effect of land processes on the current Arctic climate.

This study specifically examined a possible role of the memory effect arising from land processes, with the working hypothesis in which the negative AO–like pattern activated by Arctic sea ice loss causes significant anomalies at the surface level that are memorized in snow and soil and can persist for several months or more than a year. The memorized anomalies in turn re-emerge at the surface in later years, affecting the atmosphere and amplifying the atmospheric response related to Arctic sea ice loss. To examine this hypothesis and quantify the contribution of the memory effect, we developed a procedure to extract the memory effect in the atmospheric general circulation model (AGCM) framework (Fig. 1). As described in the Methods section, the memory effect can be estimated by the combined use of perpetual multiyear integration (a serial run) and a multi-time 1-year integration (an initialized run) of the AGCM forced by specified seasonal-cycle boundary conditions (Table 1). Although each

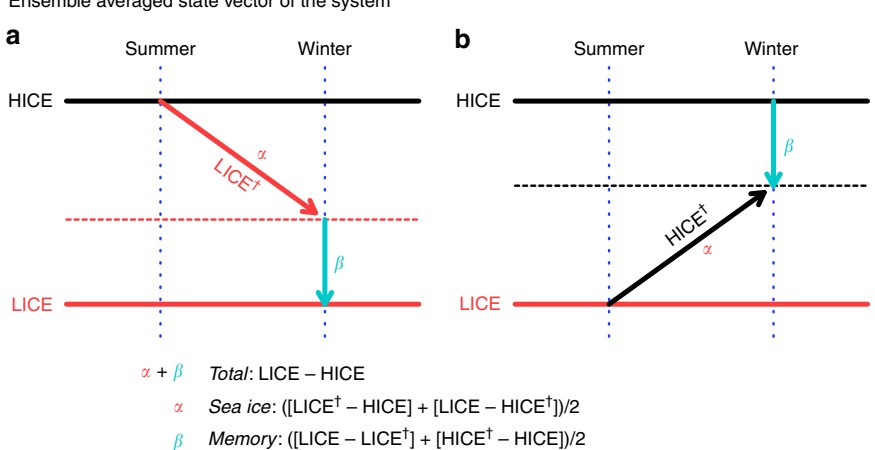

Ensemble averaged state vector of the system

α + β    *Total*: LICE − HICE
α        *Sea ice*: ([LICE† − HICE] + [LICE − HICE†])/2
β        *Memory*: ([LICE − LICE†] + [HICE† − HICE])/2

**Fig. 1** A schematic diagram of the experimental design to estimate the memory effect. Hundred-year averaged state vectors of the system. Systems with the same initial condition but different boundary conditions of high and low ice years immediately separate and become stable around individual states suitable for high and low ice conditions. Here, these states are defined as HICE and LICE, and indicated by black and red solid lines, respectively. **a** A state with a low ice condition initialized by atmospheric and land conditions existing on 1 July during HICE integration will tend to converge to the LICE state. However, during development of the atmospheric state from one summer to the next winter timeframe, it will only reach an intermediate state. This state is defined as LICE†, indicated by red arrow. The estimated memory effect is defined as the residuals of the LICE and LICE† states. **b** Similarly, a state with a high ice condition initialized by LICE will reach another state defined as HICE† and indicated by black arrow. The averages of the estimated values from both LICE† and HICE† were used for analyses according to the formulations shown under the panels (see the Methods section)

**Table 1 Brief summary of the experimental setting**

| Run | Integration | Sea ice boundary condition | Initial condition |
|---|---|---|---|
| HICE | Serial, 100 years | 1979–1983 average | 10-year spin-up |
| LICE | | 2005–2009 average | |
| HICE† | 100 iterations of 1 year | Same as HICE | Every 1 July of LICE |
| LICE† | | Same as LICE | Every 1 July of HICE |

The SST boundary condition is commonly defined as the climatological mean of the 1981–2010 period for all runs

method, serial and initialized integration, has been used in many previous studies, their combined use is a novel design that makes quantitative evaluation of the memory effect of land processes possible. We applied this procedure in a sea ice sensitivity experiment using an atmospheric general circulation model, AGCM for Earth Simulator (AFES) version 4.1, which has successfully reproduced observed sea ice impacts on the winter climate[22,23].

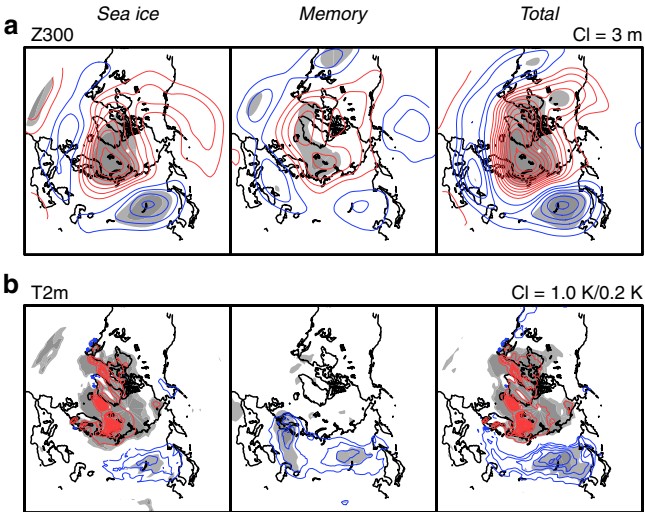

**Fig. 2** Simulated winter atmospheric responses to sea ice reduction. Winter (December–February) averaged anomalies of **a** geopotential height at 300 hPa (Z300) with a contour interval (CI) of 3 m and **b** temperature at 2 m height (T2m) with a contour interval of 1.0 K for positive anomalies and 0.2 K for negative anomalies. From left to right, 100-year averages of sea ice, memory, and total effects are shown. Red and blue indicate positive and negative anomalies, respectively, and light and heavy gray shading indicate statistical significance exceeding 95 and 99%, respectively

## Results

**Amplified Arctic–mid-latitude linkage by memory effects.** First, we examined the winter (December–February) averaged anomalies in the Northern Hemisphere. In the upper tropospheric geopotential height field (Z300), the sea ice effect from Arctic sea ice loss appears as positive anomalies over the Arctic Ocean and negative anomalies over East Asia, a combination that resembles the negative phase of AO (Fig. 2a, sea ice). This sea ice effect explains more than one-half of the total effect (total), whereas the memory effect has a similar negative AO–like pattern, with slightly smaller amplitudes (memory). The pattern in the near-surface air temperature (T2m) anomaly from the sea ice effect corresponds well with the Z300 anomaly pattern, that is, warm anomalies over the Arctic Ocean and cold anomalies over East Asia (Fig. 2b, sea ice). Naturally, no anomaly emerges from the memory effect over the Arctic Ocean while cold anomalies appear over large parts of Eurasia (memory).

It should be noted that the responses are strongly nonlinear (Supplementary Note 1). Manifestation of the negative AO-like and Eurasian mid-latitude cooling anomalies are strongly dependent on the land conditions adjusted to low sea ice conditions (Supplementary Fig. 1). Although it is unclear why such nonlinearity occurs, the atmosphere and land conditions adjusted to the high sea ice condition are more stable with respect to the sea ice change than those adjusted to the low ice condition. To shift the climate regime from one adjusted to the high ice condition to one adjusted to the low ice condition might require repeated forcing from persistently low sea ice conditions, as have been observed in recent decades.

The annual evolution of the polar cap height anomaly at 300 hPa is similar in both the sea ice and memory effects, and positive anomalies peak in winter (Fig. 3a). Their amplitude is comparable, but slightly less in the memory effect. In contrast, the near-surface polar cap temperature anomaly shows a clear difference between the two effects, the sea ice effect showing the development of a warm anomaly in winter, but the memory effect showing no such signal (Fig. 3b). The continental mid-latitude temperature anomaly also shows a difference, in which

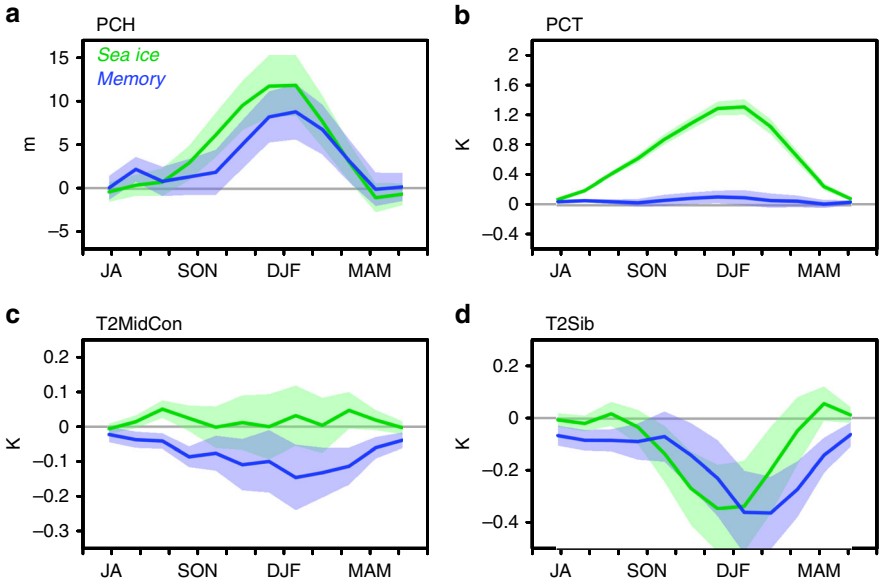

**Fig. 3** Comparison of polar amplification indicators for seasonal and memory effects. Annual evolutions of 3-month mean anomalies of **a** polar cap height (PCH) at 300 hPa, **b** polar cap temperature (PCT) at 2 m height, **c** continental 2 m temperature averaged over 40–60°N latitude (T2MidCon), and **d** 2 m temperature averaged over the Siberian region (90–140°E and 40–60°N) (T2Sib). Polar cap is defined as northward of 65°N. Green and blue indicate anomalies of sea ice and memory effects, respectively. Shading indicates the denominator of the $t$ value ($= \sigma / N^{1/2}$)

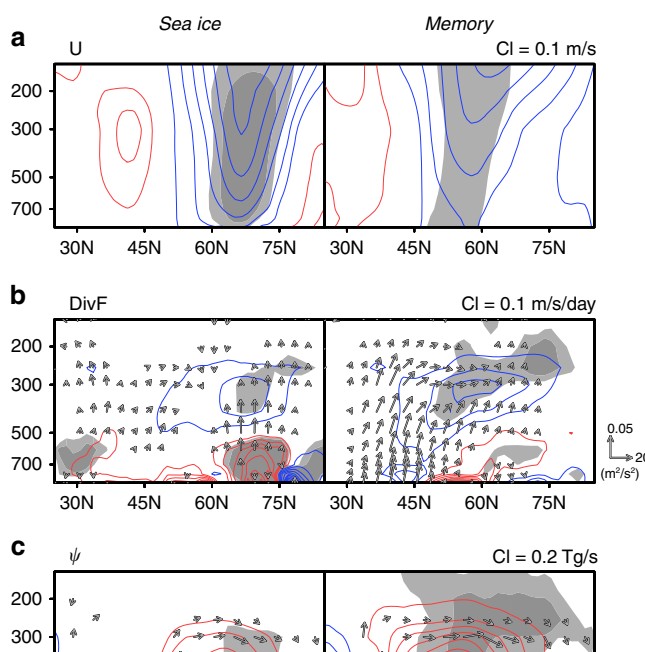

**Fig. 4** Transformed Eulerian mean diagnosis for seasonal and memory effects. Winter (December–February) averaged anomalies of global zonal mean **a** zonal wind (U), **b** zonal acceleration due to wave forcing (DivF), and **c** the mass stream function of residual mean circulation (ψ). Arrows in **b** and **c** indicate wave activity flux (EP flux) and residual mean flow vector, in which the vertical component is multiplied by a factor of 400. A legend for vector magnitude is shown at the far right of each panel. Left and right panels show 100-year averages for the sea ice and memory effects, respectively. Red and blue indicate positive and negative anomalies, respectively, and the contour interval (CI) is indicated at the top-right corner of each panel. Light and heavy gray shading indicate statistical significance exceeding 95 and 99%, respectively

**Table 2 Winter (December–February) averaged column-heating rate (W m$^{-2}$) due to the atmospheric heat transport and turbulent heat flux anomalies**

|  | Mid-latitudes[a] (Atm) | High latitudes[b] (Atm) | High latitudes (Ice) |
|---|---|---|---|
| Sea ice | −0.41 | 0.77 | 2.86 |
| Memory | −0.35 | 0.92 | −0.24 |
| Total | −0.76 | 1.69 | 2.62 |

Atm indicates mass weighted vertical integration of heat transport from 850 to 300 hPa; ice indicates turbulent heat flux from the surface to the atmosphere
[a]Area-weighted average for 30–60°N
[b]Area-weighted average for 60–90°N

the sea ice effect yields no significant anomaly, but the memory effect yields a cold anomaly (Fig. 3c). Focusing on the mid-latitude temperature anomaly over the Siberian/East Asian region, the memory effect has a cold anomaly during mid-winter, and the sea ice effect has a similar anomaly a month earlier (Fig. 3d).

Naturally as a result of our experimental design, the sea ice effect contributes much to the near-surface warming over the Arctic Ocean. Likewise, the sea ice effect contributes to Siberian/East Asian cooling in winter, which is consistent with a known feature of the sea ice–mid-latitude climate linkage[24,25]. Yet, it is surprising that the memory effect induces a negative AO–like pattern with a comparable magnitude to the sea ice effect, despite the fact that surface anomalies are present only over land.

**General circulation anomalies on the mean meridional plane.** Considering the steady-state response, a rise in the polar cap height in the upper tropospheric level (Fig. 3a) is an indicator of an increase of atmospheric column heat over the polar cap. Two main drivers increase heat in the tropospheric column. One is the direct/local impact of sea ice loss, namely the additional heat flux from the surface to the atmosphere (i.e., turbulent heat flux). The other is the indirect/remote impact, that is, changes in atmospheric heat transport due to atmospheric circulation in response

to sea ice loss. Because the memory effect has no external heat source, heat transport due to anomalous atmospheric circulation is presumably responsible for its contribution to polar heating. Here, we examine changes in atmospheric circulation on the meridional plane, which is an indicator of the atmospheric heat transport process, based on a transformed Eulerian mean (TEM) diagnosis (Methods).

In both the sea ice and memory effects, circumpolar zonal winds are weakened, which is a feature consistent with negative AO–like anomalies (Fig. 4a). This weakening is caused by wave-forcing anomalies resulting from the convergence of wave activity in the upper tropospheric level (Fig. 4b). However, the main source regions of the wave activity flux are in the mid-latitudes (around 40°N) for the memory effect and at the latitude of the Arctic coast (around 70°N) for the sea ice effect. In spite of these different pathways of wave activity, the anomalous secondary circulations generated in the meridional plane are similar (Fig. 4c), because the wave-forcing action takes place at similar locations. It is important to point out that the anomalous circulation in both effects corresponds to enhanced atmospheric heat transport from the mid-latitudes to the polar region.

We also compared the column-wise heating rate caused by atmospheric heat transport with the turbulent heat flux anomaly over the Arctic (Table 2). The average winter (December–February) turbulent heat flux anomaly due directly to sea ice loss warms the atmosphere over the Arctic region by 2.62 W m$^{-2}$. In addition to the turbulent heat flux, the anomalous meridional circulation also warms the Arctic atmosphere by 1.69 W m$^{-2}$. This additional warming by dynamical positive feedback is composed of 0.77 and 0.92 W m$^{-2}$ in sea ice and memory effects, respectively, which means that the Arctic region receives an additional 27% (i.e., 0.92/[2.62 + 0.77]) of heating through the memory effect arising from land processes. The anomalous meridional circulation also acts to cool the mid-latitudes by −0.76 W m$^{-2}$, in which the sea ice and memory effects contribute −0.41 and −0.35 W m$^{-2}$, respectively.

**Roles of Eurasian land processes for the memory effect.** As described in the previous sections, anomalous wave activity induced by the memory effect appears to originate from the mid-latitudes (Figs. 2b, 4b). Here, we examine the roles of land processes in the simulated memory effect that intensifies Arctic warming signals throughout the winter.

Positive soil temperature anomalies associated with the sea ice effect appear at the latitude of the Arctic coast, with the most rapid development in late summer (Fig. 5a, sea ice). An associated negative anomaly of snow cover is found over the Arctic coast (Fig. 5b, sea ice). Positive snow cover anomalies also appear from autumn to winter over the mid-latitudes, which is consistent with the known sea ice–snow cover relationship[1,26,27]. The memory effect in soil temperature produces persistent cold anomalies over

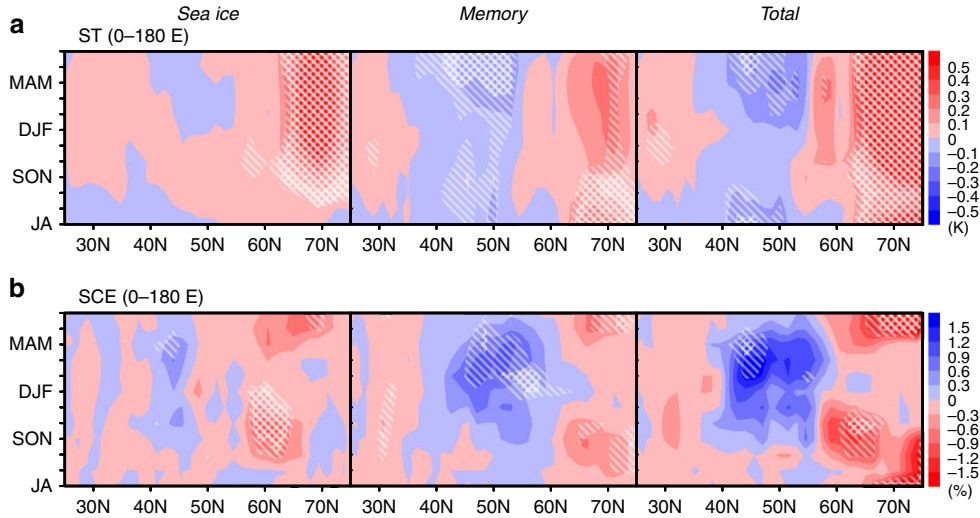

**Fig. 5** Anomalies of land process over the Eurasian continent. Annual evolution of 3-month mean anomalies of **a** soil temperature (ST) average from the surface to 4 m depth and **b** snow cover extent (SCE) over the Eurasian continent (0–180°E). From left to the right, panels show 100-year averages of sea ice, memory, and total effects, respectively. Red and blue colors are defined in the legend at the right of each panel. Hatched and double-hatched areas indicate statistical significance exceeding 95 and 99%, respectively

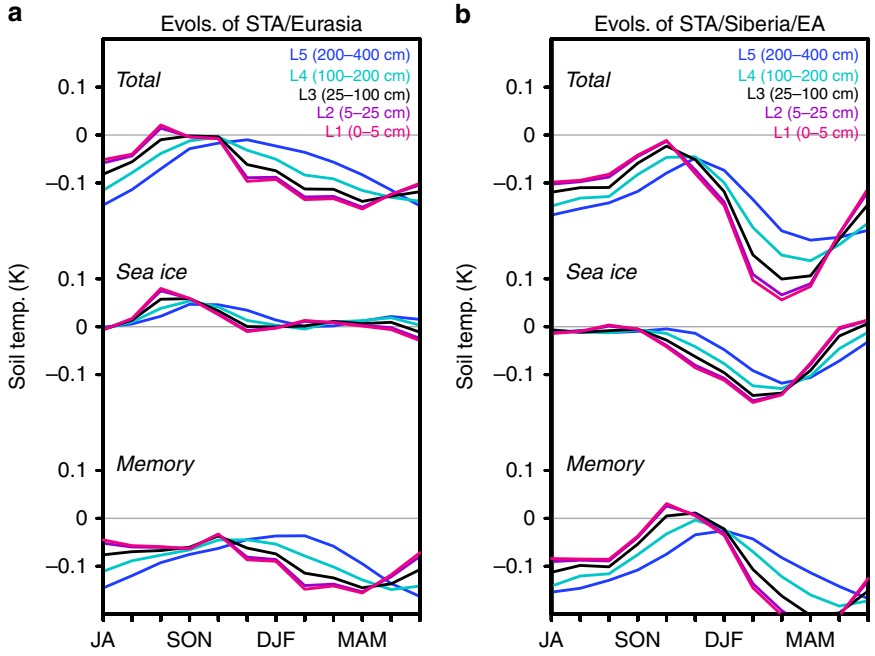

**Fig. 6** Annual cycle of the soil temperature anomalies in the model's soil layers. Seasonal evolutions of soil temperature anomalies (STA) in the five individual soil layers (L1–L5) averaged over **a** mid-latitude Eurasia (30–140°E, 40–60°N) and **b** Siberia/East Asia (90–120°E, 40–60°N), respectively. From top to the bottom, anomalies of total, sea ice, and memory effects are shown

the Eurasian continent that weaken in the summer to autumn and strengthen in the late winter (Fig. 5a, memory). Whereas the soil temperature anomaly disappears in autumn, a positive snow cover anomaly appears in autumn (Fig. 5b, memory). It then extends toward the lower latitudes in mid-winter and shrinks back toward the higher latitudes in late spring, in accordance with the climatological seasonal cycle of snow cover extent. These memory effects of land processes are dominant, especially in the mid-latitudes, where cold and increased snow anomalies persist. Thus, the memory effects make a large contribution to the total effect (Fig. 5a, b, total).

We also examined a simulated annual cycle of soil temperature anomalies in the five individual soil layers. Soil temperature

anomalies of total effect show a reasonable annual cycle corresponding to the memorized coldness. In the continental-wide land condition, near-surface layers first become cold in the winter season and then deeper layers become cold, with a peak cold anomaly later in summer (Fig. 6a). The sea ice effect appears to have no anomaly corresponding to the coldness. Rather, this annual cycle is dominantly caused by the memory effect, which shows a clear annual cycle with a half-year lag between the near-surface and bottom layers. In summer to autumn, cold anomalies in the deeper layers gradually reduce their amplitude while cold anomalies in the near-surface layers are maintained. This would bring cold surface air temperatures and early snowfall/snow cover in autumn (Fig. 5), indicating feedback of memorized coldness

beyond the annual cycle. In the Siberian/East Asian land condition, this type of annual cycle is more obvious (Fig. 6b). Furthermore, in this region, the sea ice effect has a significant role in activating the coldness. Near-surface soil temperature anomalies have a cold peak in winter and become zero in summer, while deep layers retain some extent of the cold anomalies in summer. Overall, coldness in the soil layers is forced by the sea ice effect every year, and thus remains as "cold memory" beyond the annual cycle (see also Supplementary Note 2 and Supplementary Fig. 2).

The atmospheric general circulation changes resulting from the memory effect of land processes can be explained as follows. During the cold season, the emergence of an increased snow cover anomaly (Fig. 5b) and its associated surface cold anomaly over Eurasia (Fig. 2b) induce anomalous wave activity in the mid-latitudes that enhances heat transport into the Arctic region. During the warm season, the cold anomaly is memorized in the soil and thus is not removed as readily by internal atmospheric variations. The effect of this cold anomaly, operating through the Eurasian land process, has significant impacts—comparable with the direct impacts of sea ice changes—on accelerating the Arctic warming.

## Discussion

Our innovative numerical experiment provides supporting evidence for the climate memory effect arising from Eurasian land processes. Like the oceanic re-emergence process, in which temperature anomalies are hidden in deep parts of the ocean mixed layer in summer and re-emerge in winter[28,29], temperature anomalies are stored deep in the soil layer and re-emerge at the surface in the cold season. It is essential to recognize that this memory mechanism, triggered by sea ice-induced surface cooling over Eurasia, enhances a response resembling the negative AO phase. This positive feedback, through meridional secondary circulation, warms the Arctic atmospheric column by about an additional 27% relative to the direct turbulent heat flux anomalies from sea ice loss and thereby further accelerates Arctic warming. Land surface conditions have a surprisingly large memory, and potentially an important impact on Arctic circulation, and thus the Arctic–mid-latitude linkage. The treatment of land surface conditions is important in climate models. Because the simulated land surface would be a large source of uncertainty among models, it deserves more attention.

There are several implications for uncertainties in Arctic climate simulations. First, land processes, whose roles in Arctic–mid-latitude linkage are less well understood than those of other processes, such as SST, sea ice, and atmospheric dynamics, are possible factors causing uncertainty in simulations of Arctic–mid-latitude linkage under current climate conditions and in future projections. Certainly, any future discussions of Arctic amplification need to take into account the feedbacks between the land, sea ice, and atmosphere, as well as other types of feedbacks. Furthermore, based on our results, we speculate that part of the uncertainty is due to experimental design, especially the integration methodology—for example, whether a repeated annual cycle or a historical variation is given as a boundary condition—which might suppress or amplify the memory effect from land processes.

Another implication concerns the potential roles of land processes on mid-latitude climate and weather. In this study, we focused on the Arctic–mid-latitude linkage driven by Arctic sea ice changes. However, many other factors could affect Eurasian soil conditions, such as tropical SST variations, and internal atmospheric variations. It is probable that the impacts of a large climatic event (e.g., El Niño/La Niña) are stored in the soil as a memorized anomaly that may modulate the climate several years after the initial impact. In experiments such as AMIP-type simulations, these types of extra sea ice impacts might disturb sea ice-induced memories (see Supplementary Note 3 and Supplementary Fig. 3).

This study has some limitations. Consistent with our simulation, observations of Eurasian snow anomalies show an increase of snow amount during permanent snow cover seasons with a decreasing duration of snow days, especially in spring[1,30]. On the other hand, the Eurasian continent, especially northern Eurasia, has experienced severely hot summers since the beginning of this century[31,32]. Such surface warming might have suppressed the re-emergence of cold anomalies in the soil layer. In situ observations of Eurasian deep soil conditions are sparse, especially in the mid-latitude region where cold signals are strongest in our simulation (see Supplementary Note 4 and Supplementary Fig. 4). An intensive observation campaign and reconstruction of past records would be helpful to quantify the memory effect of land processes with more confidence.

Multi-model intercomparisons are currently being conducted to evaluate the Arctic amplification occurring in the current climate and in future projections (e.g., the Polar Amplification Model Intercomparison Project, PAMIP[33]). Insights gained from this study with regard to land processes and their role in climate memory can be used to improve the design and interpretation of the results of future studies on Arctic climate change.

## Methods

**Data and experimental setup.** We used AFES version 4.1, with a horizontal truncation of T79 (horizontal resolution of approximately 150 km) and 56 vertical levels up to the 60 km model top. AFES incorporates a land surface model, minimal advanced treatments of surface interaction and runoff (MATSIRO)[34], to simulate ground processes, including soil temperature and snow depth. Daily mean output is used for analysis. To estimate the memory effect of land processes, we performed two numerical simulation experiments that used monthly mean data from the Merged Hadley–National Oceanic and Atmospheric Administration (NOAA)/ Optimum interpolation (OI) SST data set[35] as the boundary condition (briefly summarized in Table 1). The first simulation was a serial time-slice experiment performed with high and low sea ice boundary conditions (HICE and LICE runs, Fig. 1) and climatological mean SST. High and low sea ice years were defined as the 5-year averages of the 1979–1983 and 2005–2009 periods, respectively, and the SST climatological mean was defined as the 30-year average of the 1981–2010 period. This is the same setup as that used by our previous study[36], which provides additional details regarding other external conditions. A 100-year period after a 10-year spin-up was used for all analyses. The second simulation was an initialized experiment where 100 iterations of a 1-year integration were run (LICE† and HICE† runs, Fig. 1). The LICE† run was performed with the LICE boundary condition, but it was initialized every year on 1 July using the land and atmospheric fields from HICE (Fig. 1a). Similarly, the HICE† run was performed with the HICE boundary condition, but it was initialized by the LICE run (Fig. 1b).

Using 100-year average outputs from each run, the atmospheric response to the total effect of Arctic sea ice loss was estimated as (LICE–HICE) (total). The atmospheric response that develops during summer to the following winter as a direct effect of sea ice loss was estimated as ([LICE†–HICE] + [LICE–HICE†])/2 (sea ice). If the anomalies due to the sea ice effect explain most of the total effect, the LICE† and HICE† states should be close to the LICE and HICE states, respectively, indicating there is no memory effect beyond the annual cycle. Therefore, the residual of the total minus the seasonal effects is regarded as the memory effect that persists from previous years (memory). In the proposed framework, because the atmospheric internal variation generally does not persist for longer than a few weeks, the memory effect is achieved only through land processes. Thus, this experimental design is suitable for the purpose of this study.

**Statistics and techniques.** AFES involves five ground layers down to a 4 m depth for simulating soil temperature and moisture. The soil data were analyzed as the weighted vertical average, considering the thickness of each layer. The transformed Eulerian mean (TEM) applied to the daily mean dynamical field was used to diagnose changes in the atmospheric general circulation in the meridional plane, which is indicative of heat transport from the mid-latitudes to the polar region[8,36]. A column-heating rate due to atmospheric heat transport was obtained from the vertical velocity in the TEM framework. The monthly mean snow cover extent was calculated from the daily mean snow mass output as a fraction of the number of days when the snow depth was >2 cm in a grid cell during a month[37]. To eliminate high-frequency noise caused by atmospheric internal variability, a 3-month

running mean was applied to all variables before the analysis. For the 100-year integration output as an ensemble member, we examined total, sea ice, and memory effects using 100-year averages of individual runs. Statistical significances were evaluated by using a two-tailed standard *t* test for 100 samples.

## Data availability
The AGCM simulation data used in this study are available from the corresponding author upon reasonable request.

## Code availability
All codes used for analyses of the simulation data are available from the corresponding author upon reasonable request.

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

## Acknowledgements
The Merged Hadley–NOAA/OI SST and SIC data were obtained from the Climate Data Guide (https://climatedataguide.ucar.edu/). The simulations were performed on the Earth Simulator at the Japan Agency for Marine-Earth Science and Technology. To access our simulation data, contact the corresponding author (nakamura.tetsu@ees.hokudai.ac.jp). We are grateful for support from the Arctic Challenge for Sustainability project and the InterDec project of Belmont Forum. This study is supported by MEXT KAKENHI grant no. 18K0373508.

## Author contributions
T.N. designed the research, conducted the numerical experiment and the analysis, and wrote the majority of the paper. K.Y., T.S., and J.U. supported the analysis. All authors discussed the study results and reviewed the paper.

## Competing interests
The authors declare no competing interests.
