## [Peer Review File · Nature Communications]

Reviewers' comments:

Reviewer #1 (Remarks to the Author):

The article "Climate memory of Eurasian land processes for Arctic amplification" by Tetsu Nakamura, Koji Yamazaki, Tomonori Sato, Jinro Ukita provides interesting new results on the potential impact of the soil processes and their memory on Arctic warming and its potential feedback on lower latitudes. The authors performed model sensitivity experiment to investigate the importance of initial land surface conditions for Arctic climate. The results suggest that land surface processes are an important contributor to Arctic warming and feedbacks on lower latitudes and suggest a positive feedback between mid-latitude soil temperature, snow and Arctic warming. The study thus showed the importance of a correct representation of the land surface of models for simulating the Arctic amplification and its potential feedbacks .

While the method is solid and the results new and interesting and thus worth to publish, I have some smaller concerns about the context, this work is set into. The model experiments are highly idealized (which is absolutely fine) and should be treated as such in the discussion, and the results from these highly idealized experiments should be interpreted carefully when comparing to the real world.

The authors suggest that land-surface processes could explain why different models show contrasting results on the impact of sea ice reductions and variations on lower latitudes. This might be true for some idealized experiments with e.g. repeating annual forcing data. However, it is not real true for the large number of AMIP-type experiments forced with observations from 1980 (or similar)-today (unless land surface schemes are all very uncertain, which in this context would mean that also results from this study are very uncertain).

Further, HICE and LICE conditions are calculated by using averages over several years as forcing. The LICE* runs are started from HICE-land surface conditions and vice versa. The land surface conditions in idealized 100-year serial HICE/ LICE experiments are very likely not very realistic compared to real world land surface conditions in a real HICE/ LICE year. Jumps from HICE land surface conditions (1979-1983) directly to low ice sea ice forcing (2005-2009) will normally not happen.

These idealizations in the experiments makes it impossible to link the results directly to the real world. The conclusions should thus focus mainly on: Land surface conditions have a surprisingly large memory and potentially an important impact on Arctic circulation. They potentially have an important effect on sea ice – lower latitude interactions. The treatment of land surface conditions is highly important in climate models, and since the land surface is still rather primitively simulated in most models, it deserves more attention.

Some minor comments:

Title: To me the title sounds a bit strange, I suggest to reformulate it.

Line 13: "consistent observational evidence showing strong connections". I only partly agree: Firstly, the term "mid-latitude cooling" is not really correct. Mid-latitudes are only cooling in certain regions; as a whole they are warming (at least when looking at somewhat longer periods). The only region where a stronger cooling is found are parts of Asia/ Siberia. But even here, e.g. Ogawa et al. (2018) shows that the observed cooling trend is only in very few grid points significant at the 95% level between 1982-2014. Winters after 2014 were warmer again, which reduced the cooling trend and further reduced the significance of this trend. Only if one starts in year 1990 (or years around), exactly in the period with the most extreme positive NAO-years in the last 50 years or more, a real strongly significant cooling trend occurs together with a negative NAO-trend. Further, the strength of the linkage between sea ice loss and the cooling depends rather strongly on the period that is investigated.

L28: Arctic amplification is normally defined as a "larger warming of the Arctic compared to the global mean warming" not as the "acceleration of warming over the Arctic". I suggest, that the authors stick to this definition.

L31ff: In this discussion, it should be noted that there is no negative trend of the NAO anymore. Recent winters showed mostly positive NAO-indices, which makes the trend between 1980-2018 being zero. Thus, the trend of the NAO can not any longer that easily be connected to the negative

sea ice trend. Sea ice loss might still affect the NAO but then other processes are compensating this sea ice effect on the NAO.

Line 63: "in later years". I am not sure that the soil keeps the memory over several years.

Figure 1/ Figure 2: The memory effect should also be calculated separately for LICE*-LICE and HICE*-HICE. The effect could strongly depend on the initial land surface conditions (HICE or LICE).

Line 79: I would suggest to change this sub-title as well. There is hardly any analysis of the "Arctic Amplification" in this section.

Line 157-158: What causes the snow cover anomalies? Does the "memory" show any changes in the atmospheric circulation in autumn that could explain the increased snow?

Line 176: The soil temperature anomaly should have a seasonal cycle in depth. It should be assessed in which level/ depth the temperature signal is stored during summer.

Torben Koenigk

Review of Nature Communications manuscript # NCOMMS-19-05429

Climate memory of Eurasian land processes for Arctic amplification

by Nakamura, Yamazaki, Sato, and Ukita

Overview

This study addresses an interesting topic: the role of soil and snow-covered land in creating a memory mechanism in the climate system that augments Arctic amplification, which in turn, affects the large-scale atmospheric circulation. The authors use a clever combination of model simulations to isolate the land-originating effects from those related purely to sea-ice loss (though admittedly it took a while to get my head around the approach and I'm not confident that I was 100% successful!). They contrast two sets of model simulations. The first (called seasonal runs) is forced only by a repeating annual cycle of average sea-ice extent and SSTs in years with either low or high sea-ice extent. The second set (memory runs) begins with boundary conditions for low (high) sea-ice extents (same as first set), but on every July 1st of each 100-year run, the land and atmosphere conditions corresponding to high (low) sea-ice years are imposed, after which the atmosphere evolves accordingly. Because sea ice and SST states are the same in both sets of simulations, any differences between the atmospheric responses are interpreted to be caused by processes involving the land, i.e., soil temperatures and snowcover.

Their results suggest that low summer sea-ice leads to cooling over Eurasia owing mainly to the memory effect, which causes anomalously cold soil temperatures and an increased snowcover extent in mid-latitudes. The cold anomaly weakens in late fall/early winter, then emerges strongly in spring owing to the soil's ability to "memorize" the cold anomaly (soil layer from surface to 4m depth). The cold anomaly then leads to anomalous wave activity, which weakens zonal winds north of about 50°N, increases poleward heat transport, and augments warming over the polar cap along with a negative Arctic oscillation.

While I found this study novel, interesting, timely, and potentially important, I have two major concerns. First, the results hinge on the ability of their model to realistically simulate energy transfer within the top 4 m of soil in a region riddled with complex terrestrial conditions, such as permafrost, saturated bogs, and complex geology. The authors do mention that land processes in climate models are less well understood and that observations for ground-truthing (sorry for pun) are sparse, but some evidence for the model's ability to capture gross temperature profiles within the terrestrial system should be provided. Bore hole temperatures used to monitor permafrost degradation may be helpful. No references are cited regarding the simulated soil processes. My second concern is that the relationship between soil temperature and snowcover seems to be opposite to basic physics and observations (Fig. 5). Generally a layer of snow acts as insulation for the surface below (be it ice or soil), yet their results suggest that as snow extent increases anomalously over Eurasia, negative soil temperature anomalies intensify. Moreover, spring snowcover on high-latitude land areas has been declining

dramatically in recent decades, which seems opposite to results from the memory experiment. Perhaps I am misinterpreting the message from these results; if so, perhaps some clarification is needed in the text regarding these points. Given these potentially major concerns, I recommend that the manuscript undergo revision before acceptance. Additional analyses may be necessary, which could be included in supplemental material.

Specific comments

1. Line 121: add “(Methods)” after TEM diagnosis
2. 147: Sentence refers to Eurasia, but Fig. 4b is for all longitudes, correct? If not, caption for Fig. 4b should be clarified.
3. 153: This statement needs clarification. I suggest “Increased negative snowcover anomalies...”.
4. 157: It looks like the anomaly weakens in autumn rather than summer – easier to see this in Fig. 3d.
5. 160: insert “late” before spring?
6. 161: How does this result relate to the fact that spring snowcover has been declining in recent decades?
7. 167: Perhaps “disturbed” should be replaced by “removed as readily”?
8. 176: Please add a reference describing the re-emergence of ocean temperature anomalies in winter.
9. 179: suggest removing “as a potential cause”
10. 181: suggest clarifying this statement by adding “turbulent-flux” before heating
11. 195: suggest removing “positive” (maybe negative ones, too) and make feedback plural – probably more than one feedback process.
12. 196: make feedback plural
13. 198-199: suggest rewording, such as: “...whether or not a repeated annual cycle is a realistic boundary condition – which might suppress or amplify the memory effect...”
14. 206: change “would” to “could”?
15. 209: Clarify “snow days” – does this mean the number of days in the year with snowcover or days when snow falls?
16. 226: define AFES. What is temporal resolution of output from simulations?
17. Fig. 1a: Are the initial atmospheric and land conditions for both serial experiments consistent with HICE? That is what the diagram implies. If so, perhaps the caption should include this detail.
18. Fig. 1 and throughout text: The label “seasonal” for one of the initialized experiments seems somewhat confusing, as there are seasons in both sets of experiments. Perhaps consider calling it “sea ice” to distinguish it from the other experiments that isolate the land effects? I suggest using shading in the row labeled as memory effect to help delineate these experiments. Also please label which sections of the schematic represent “total”. The caption for this figure could be a bit clearer, in my view. Following are some suggestions to help with that.

19. 393: before "HICE" add "atmospheric and land conditions existing at July 1st during"
20. 395: add "of the atmospheric state" after "development"
21. Fig. 3 caption: add (PCH) after polar cap height and (PCT) after polar cap temperature.
22. Fig. 4: Are these results only for Eurasia or for all longitudes? Please make a note in the caption, just to be clear.
23. Fig. 5: I think it would be very helpful to add another set of plots for the "Total" response. This would help understand how the seasonal and memory responses contribute to the total response. Hatch marks are very difficult to see – perhaps try white or yellow instead?

Reviewer #3 (Remarks to the Author):

The manuscript titled "Climate memory of Eurasian land processes for Arctic amplification" explores the role of Eurasian land processes in the Arctic warming and mid-latitude cooling by conducting a set of numerical experiments. Results from the serial experiments give full response of climate to the forcing of sea ice reduction, i.e. direct response to sea ice reduction plus feedbacks induced by the direct forcing. On the other hand, ensemble initialized experiments estimate the response of climate to sea ice reduction in the first year, which covers the fast component with a time scale less than a year and thus is assumed to be the direct response. The difference between the fast response and the full response is mainly due to the slow land processes and is attributed to the land memory effect by the authors. While this is a nice approach to separate the fast and slow components of the responses, a couple of issues need to be addressed.

1. The major concern from the reviewer is how does the memory component evolves on the inter-annual time scale. Since the authors define the memory component as the difference between the full response and the first year response, the we can only see the seasonal cycle of the memory component. However, it is also important to know when the memory component starts to develop and peak, its year-to-year evolution and its time scale character.

2. My second comment involves a couple of questions. Can we test the modeling results by comparing to observations? Can we detect the land memory component from observational data? Or, can we look at other model output (e.g. CMIP5) and see if they support the findings in this study or are just again what have been found here? This would make the results more relevant and robust.

Reply to Reviewer 1 (Dr. Torben Koenigk):

Dear Dr. Koenigk,

Thank you very much for your constructive remarks on our original manuscript. We have replied to your comments below and outlined the changes made to the revised manuscript. Please note that we changed from the use of the term “seasonal” effect, which referred to the half-year response to the Arctic sea ice loss, to the “sea ice” effect throughout the entire main text and supplementary files.

Reviewer #1 (Remarks to the Author):

The article “Climate memory of Eurasian land processes for Arctic amplification” by Tetsu Nakamura, Koji Yamazaki, Tomonori Sato, Jinro Ukita provides interesting new results on the potential impact of the soil processes and their memory on Arctic warming and its potential feedback on lower latitudes. The authors performed model sensitivity experiment to investigate the importance of initial land surface conditions for Arctic climate. The results suggest that land surface processes are an important contributor to Arctic warming and feedbacks on lower latitudes and suggest a positive feedback between mid-latitude soil temperature, snow and Arctic warming. The study thus showed the importance of a correct representation of the land surface of models for simulating the Arctic amplification and its potential feedbacks .

While the method is solid and the results new and interesting and thus worth to publish, I have some smaller concerns about the context, this work is set into. The model experiments are highly idealized (which is absolutely fine) and should be treated as such in the discussion, and the results from these highly idealized experiments should be interpreted carefully when comparing to the real world.

The authors suggest that land-surface processes could explain why different models show contrasting results on the impact of sea ice reductions and variations on lower latitudes. This might be true for some idealized experiments with e.g. repeating annual forcing data. However, it is not real true for the large number of AMIP-type experiments forced with observations from 1980 (or similar)-today (unless land surface schemes are all very uncertain, which in this context would mean that also results from this study are very uncertain).

Thank you for your comments. In AMIP-type simulations, sea ice forcing does not necessarily repeat, but rather, it varies year to year. Even in this context, however, we think the memory effect of the land process presented in this study is still important. As we discuss in the Discussion and Supplementary S2, the preceding summer-to-autumn land surface condition has the potential to affect the following winter climate even in an AMIP-type simulation. However, in an AMIP-type simulation, various types of forcing occur not only for sea ice but also for SST variations, radiative forcing due to increasing GHGs, stratospheric conditions such as the QBO, ozone variations, and tropical intra-seasonal variation such as the MJO. Our AMIP-type simulation suggested that such extra sea ice forcing would have impacts on land process and interrupt sea ice-induced memories (Supplementary S2).

We still do not know if such extra sea ice-related memories (e.g., ENSO-related anomalies) have constructive, destructive, or no impacts on sea ice-related anomalies, and it may vary depending on the model. Therefore, a multi-model comparison with a focus on the memory effect would be a useful next step.

Further, HICE and LICE conditions are calculated by using averages over several years as forcing. The LICE* runs are started from HICE-land surface conditions and vice versa. The land surface conditions in idealized 100-year serial HICE/ LICE experiments are very likely not very realistic compared to real world land surface conditions in a real HICE/ LICE year. Jumps from HICE land surface conditions (1979-1983) directly to low ice sea ice forcing (2005-2009) will normally not happen.

Your concern highlights an important implication of this study. As we noted in Fig. 1 of the previous version, Deser et al. (2007), Honda et al. (2009), Mori et al. (2014) and others adopted this sudden replacement of sea ice condition. This type of replacement may be correct if the memory effect is assumed to be small.

Because we only present results from our model, we cannot determine if the sea ice-related memories are really comparable with the direct sea ice effects. Future work on a multi-model evaluation of the memory effect will be very important to examine these effects.

These idealizations in the experiments makes it impossible to link the results directly to the real world. The conclusions should thus focus mainly on: Land surface conditions have a surprisingly large memory and potentially an important impact on Arctic circulation. They potentially have an important effect on sea ice – lower latitude interactions. The treatment of land surface conditions is highly important in climate models, and since the land surface is still rather primitively simulated in most models, it deserves more attention.

Thank you for this very helpful suggestion. We agree that our results are based on highly idealized experiments, and thus, the conclusions should focus on the simulation uncertainty. We modified the main text accordingly to reduce the discussion about the real-world links to our simulation results.

Some minor comments:

Title: To me the title sounds a bit strange, I suggest to reformulate it.

We have modified the title to “The role of climate memory from Eurasian land processes in Arctic–mid-latitude climate linkage.”

Line 13: “consistent observational evidence showing strong connections”. I only partly agree: Firstly, the term "mid-latitude cooling" is not really correct. Mid-latitudes are only cooling in certain regions; as a whole they are warming (at least when looking at somewhat longer periods). The only region where a stronger cooling is found are parts of Asia/ Siberia. But even here, e.g. Ogawa et al. (2018) shows that the observed cooling trend is only in very few grid points significant at the 95% level between 1982-2014. Winters after 2014 were warmer again, which reduced the cooling trend and further reduced the significance of this trend. Only if one starts in year 1990 (or years around), exactly in the period with the most extreme positive NAO-years in the last 50 years or more, a real strongly significant cooling trend occurs together with a negative NAO-trend.

Further, the strength of the linkage between sea ice loss and the cooling depends rather strongly on the period that is investigated.

We agree that the cooling anomalies over the mid-latitudes have become weaker or changed to a warming trend in recent years, whereas Arctic sea ice has remained continuously low. The cooling trend was particularly obvious in the period between the strong positive AO/NAO in 1988/89 and the strong negative AO/NAO in 2009/10. We wanted to express the strong relationship between Arctic sea ice and mid-latitude anomalies not only on the decadal but also the interannual time scale because interannual variation still has shown a robust correlation between sea ice and mid-latitude anomalies in recent decades (Fig. R1).

However, the mid-latitude signals related to sea ice variations are the most obvious in Siberia/East Asia, and even in correlations on the interannual timescale, there is a decadal-scale change in the relationship (Fig. R1). Because the context of our study is focused on long-term changes, we should note such discussions about the long-term tendency. We modified abstract (Line 14–18) and added a sentence “*However, winter AO/NAO index has shown positive trend after the strongest negative AO/NAO year of 2010, while the Arctic sea ice stays continuously low level.*” (Line 38–40).

Figure R1. Anomalies of DJF mean temperature at 2 m height regressed on the preceding November sea ice concentration averaged over the Barents/Kara Seas ($20\text{--}70^\circ\text{E}$, $65\text{--}85^\circ\text{N}$). Contours indicate the regression coefficient at 0.5 K intervals, and shading indicates statistical significance exceeding 95%. Red and blue indicate positive and negative correlations, respectively. The regression was calculated for 20-year periods as indicated in the bottom of each panel. Variables were detrended for

individual 20-year periods before calculating the regression. The sign is reversed to correspond to sea ice decrease.

L28: Arctic amplification is normally defined as a “larger warming of the Arctic compared to the global mean warming” not as the “acceleration of warming over the Arctic”. I suggest, that the authors stick to this definition.

We have modified the text to reflect this more accurate definition of Arctic amplification.

L31ff: In this discussion, it should be noted that there is no negative trend of the NAO anymore. Recent winters showed mostly positive NAO-indices, which makes the trend between 1980-2018 being zero. Thus, the trend of the NAO can not any longer that easily be connected to the negative sea ice trend. Sea ice loss might still affect the NAO but then other processes are compensating this sea ice effect on the NAO.

We recognize that sea ice variation still is strongly related to negative AO/NAO-like patterns on an interannual timescale. However, as you note, the recent AO/NAO index has been rather positive while sea ice has been continuously low. We have modified the text to reflect this accordingly (Line 38–40).

Line 63: “in later years”. I am not sure that the soil keeps the memory over several years.

Soil memory is an important feature of this study. Although we did not specify how long the land memory persists in our model quantitatively, the memory effect evaluated by the procedure in this study must be achieved only by the land process. We performed additional experiments of “initialized runs” that extend the integration period by 3 years, and examined characteristics of the memory effect, especially how they evolve beyond interannual cycles. This new material is presented in Supplementary S3.

Figure 1/ Figure 2: The memory effect should also be calculated separately for LICE*-LICE and HICE*-HICE. The effect could strongly depend on the initial land surface conditions (HICE or LICE).

There is a strong nonlinearity in the responses between the initial land surface conditions. Strong and obvious negative AO-like circulation patterns and mid-latitude cooling anomalies are found in the memory effect with LICE's land condition (*Memory* [LICE–LICE[†]], the right panels of Fig. R2a and R2c) and in the sea ice effect with LICE's land condition (*Sea ice* [LICE[†]–HICE], the right panels of Fig. R2b and R2d). Interestingly, significant anomalies only appear when the experiments are performed with LICE's land condition. The atmosphere and land conditions that stably adjusted to the low sea ice condition appear to be more sensitive to sea ice changes as compared to those that adjusted to the high sea ice condition. Although it is unclear why such nonlinearity occurs, this information is important, so we have added these results in Supplementary S1.

Figure R2. Simulated winter atmospheric responses to sea ice reduction. This is the same as Fig. 2 in the main text except for that the *Sea ice* and *Memory* effects are separately evaluated by using **a** and **c** LICE[†] and **b** and **d** HICE[†] runs, corresponding to the left and right descriptions shown in Fig. 1, respectively. Red and blue indicate positive and negative anomalies, respectively, and light and heavy grey shadings indicate statistical significance exceeding 95% and 99%, respectively.

Line 79: I would suggest to change this sub-title as well. There is hardly any analysis of the “Arctic Amplification” in this section.

We have changed the sub-title as Arctic amplification -> Arctic–mid-latitude linkage (Line 82).

Line 157-158: What causes the snow cover anomalies? Does the “memory” show any changes in the atmospheric circulation in autumn that could explain the increased snow?

In the memory effect, the circulation anomalies do not appear to show anything that would create a snow anomaly. Rather, the cold conditions over the Eurasian mid-latitudes are brought about by soil temperature conditions, which show that cold anomalies persist from season to season. As shown in Figure R3, in summer (JAS), very small anomalies in soil temperature were found as a result of the sea ice effect, which is a reasonable result derived from the experimental procedure of this study. On the other hand, the memory effect shows cold soil temperature anomalies in summer over the mid-latitude regions where cold surface temperatures and increases in snow cover anomalies are found in the following autumn.

Figure R3. Simulated winter atmospheric responses to sea ice reduction. Anomalies of **a** temperature at a height of 2 m (T2m), **b** snow cover extent (SCE) and **c** 0–4 m soil temperature. The left and right panels are 100-year averages of the *Sea ice* and *Memory* effects, respectively. Hatch/double hatch indicates statistical significance exceeding 80/95%.

Line 176: The soil temperature anomaly should have a seasonal cycle in depth. It should be assessed in which level/ depth the temperature signal is stored during summer.

It is reasonable that the soil temperature anomaly would have a seasonal cycle in depth. Here, we show a seasonal cycle of soil temperature anomalies in the individual five soil layers of the model. The soil temperature anomalies of the total effect show a reasonable annual cycle corresponding with memorized coldness. In the continental-wide land condition, near-surface layers first become cold in the winter season, and after that, deeper layers become cold with a peak cold anomaly later in summer (Fig. R4a below, Fig. 6a in the main text). The sea ice effect seems to have no anomaly corresponding to this coldness. Rather, this annual cycle is dominantly brought

about by the memory effect, which shows a clear annual cycle, with a half-year lag between near-surface and bottom layers. The amplitude is gradually reduced in the summer to autumn cold anomalies in the deeper layers, while the cold anomalies in the near-surface layers are maintained. This situation would bring about cold surface air temperatures and early snowfall/snow cover in autumn (Fig. 5 in the main text), indicating the feedback of memorized coldness beyond seasons. In the Siberian/East Asian land condition, this annual cycle is most obvious (Fig. R4b below, Fig. 6b in the main text). Furthermore, in this region, the seasonal effect has a significant role in activating coldness. Near-surface soil temperature anomalies have a cold peak in the winter and decrease to zero in summer, while deep layers retain some extent of the cold anomalies in summer. This coldness in the soil layers is forced by the seasonal effect every year, and thus remains as “cold memory” beyond the annual cycle. Although it is unclear why or how the cold condition activated in Siberia/East Asia extends to the continental-wide regions in the memory effect, we have added this result in the main text (Fig. 6 and corresponding explanations, Line 177–194).

Figure R4. Annual cycle of the soil temperature anomalies in the model's soil layers. Seasonal evolutions of soil temperature anomalies in the individual five soil

layers (L1–L5) averaged over **a** mid-latitude Eurasia (30–140°E, 40–60°N) and **b** Siberia/East Asia (90–120°E, 40–60°N), respectively. From top to bottom, the anomalies of *Total*, *Sea ice*, and *Memory* effects are shown.

Torben Koenigk

Once again, we very much appreciate your constructive comments and suggestions. They helped us to improve the revised manuscript. We hope this revision addresses your concerns.

Reply to Reviewer 2:

Thank you very much for your constructive remarks on our original manuscript. We have replied to your comments below and outlined the changes made to the revised manuscript. Please note that we changed from the use of the word “seasonal” effect, which referred to the half-year response to the Arctic sea ice loss, to the “sea ice” effect throughout the entire main text and supplementary files.

Overview

This study addresses an interesting topic: the role of soil and snow-covered land in creating a memory mechanism in the climate system that augments Arctic amplification, which in turn, affects the large-scale atmospheric circulation. The authors use a clever combination of model simulations to isolate the land-originating effects from those related purely to sea-ice loss (though admittedly it took a while to get my head around the approach and I’m not confident that I was 100% successful!). They contrast two sets of model simulations. The first (called seasonal runs) is forced only by a repeating annual cycle of average sea-ice extent and SSTs in years with either low or high sea-ice extent. The second set (memory runs) begins with boundary conditions for low (high) sea-ice extents (same as first set), but on every July 1st of each 100-year run, the land and atmosphere conditions corresponding to high (low) sea-ice years are imposed, after which the atmosphere evolves accordingly. Because sea ice and SST states are the same in both sets of simulations, any differences between the atmospheric responses are interpreted to be caused by processes involving the land, i.e., soil temperatures and snowcover.

Their results suggest that low summer sea-ice leads to cooling over Eurasia owing mainly to the memory effect, which causes anomalously cold soil temperatures and an increased snowcover extent in mid-latitudes. The cold anomaly weakens in late fall/early winter, then emerges strongly in spring owing to the soil’s ability to “memorize” the cold anomaly (soil layer from surface to 4m depth). The cold anomaly then leads to anomalous wave activity, which weakens zonal winds north of about 50°N, increases poleward heat transport, and augments warming over the polar cap along with a negative Arctic oscillation.

While I found this study novel, interesting, timely, and potentially important, I have two major concerns.

First, the results hinge on the ability of their model to realistically simulate energy transfer within the top 4 m of soil in a region riddled with complex terrestrial conditions, such as permafrost, saturated bogs, and complex geology. The authors do mention that land processes in climate models are less well understood and that observations for ground-truthing (sorry for pun) are sparse, but some evidence for the model's ability to capture gross temperature profiles within the terrestrial system should be provided. Bore hole temperatures used to monitor permafrost degradation may be helpful. No references are cited regarding the simulated soil processes.

Thank you for this insightful comment. We agree that this is a critical point. Our simulation results suggest that sea ice reduction and the associated circulation change brought anomalous coldness over the Eurasian continent, and furthermore, the coldness persists for several years and has impacts on the atmosphere as a climate memory. However, as we note in the Discussion, a limitation of this study is the difficulty of explaining observed tendencies of ground conditions with our simulation. One of major reasons might be that the real world is undergoing global warming whereas our idealized simulation is based on repeated annual cycle of the boundary forcing.

Taking your suggestion, we conducted an additional analysis using borehole temperature data from a GTN-P dataset (Biskaborn *et al.*, 2019). Although the data period is 10 years, these *in-situ* observations are quite helpful when examining the role of ground processes. We calculated the correlation between preceding sea ice anomalies over the Barents/Kara Seas and the annual mean borehole temperatures at individual stations. The correlations were generally positive (note that sign is reversed to show anomalies associated with sea ice loss), indicating the ground temperature increase after sea ice loss is in accordance with the long-term tendency due to global warming (Fig. R1a).

On the other hand, when we looked at the relationship of interannual variations of the sea ice and the ground temperature by conducting a detrend analysis, negative correlations appear over the eastern part of Eurasia, indicating the occurrence of cold ground temperature anomalies after sea ice loss (Fig. R1b), whereas two stations near

the Barents/Kara Seas still had positive correlations. Although the correlations were not statistically significant, these signals roughly correspond to our simulation results that show a temperature response to sea ice loss (Fig. 2b and Fig. 3d in the main text).

These results are presented and discussed in Supplementary S4.

Although such observation-based analysis results roughly support our simulation results, we decided to focus on “*the simulation uncertainty*” of Arctic and mid-latitude climate linkage. This is because of the difficulties to argue consistency between observation and simulations and rather the mentioned background of this study. We think the value of this study is retained even if there is no supporting evidence from observations. The results and implications of this study have special impacts on the climate science community because researchers are interested in a model’s ability to simulate the Arctic–mid-latitude linkage, but they have generally not recognized the importance of the land process. We have carefully revised the manuscript to focus on this interpretation, and the conclusions now mainly focus on simulation uncertainty.

AFES has incorporated a land surface model, minimal advanced treatments of surface interaction and runoff (MATSIRO) (Takata *et al.*, 2003). We therefore added this reference citation, as well as the Biskaborn *et al.* (2019) reference, to the revised paper.

References

Biskaborn, B. K. *et al.* Permafrost is warming at a global scale. *Nat. Commun.* **10**, 264 (2019).

Takata, K., Emori, S. & Watanabe, T. Development of the minimal advanced treatments of surface interaction and runoff. *Glob. Planet. Change* **38**, 209–222 (2003).

Figure R1. Observation-based relationship between sea ice variation and ground temperature anomalies. Correlation coefficients between sea ice concentration over the Barents/Kara Seas (30–90°E, 65–85°N) in the previous November and annual mean bore-hall temperature data from the GTN-P dataset (Biskaborn *et al.*, 2019). All stations in the eastern Northern Hemisphere are used except stations with less than 5 years of available data or with an elevation above 3000 m. The calculation period is 10 years based on a GTN-P data period of 2007–2016. Correlation coefficients are shown for each GTN-P station calculated **a** without and **b** with detrending. Note that the sign of the coefficient is reversed to show anomalies associated with a decrease of sea ice.

My second concern is that the relationship between soil temperature and snowcover seems to be opposite to basic physics and observations (Fig. 5). Generally a layer of snow acts as insulation for the surface below (be it ice or soil), yet their results suggest that as snow extent increases anomalously over Eurasia, negative soil temperature anomalies intensify. Moreover, spring snowcover on high-latitude land areas has been declining dramatically in recent decades, which seems opposite to results from the memory experiment.

We appreciate your bringing up the concern. For the explanation below let us suppose that there is an additional amount of snow cover. Thermal insulation by snow cover acts differently during surface cooling and warming periods. In the case of surface cooling, snow cover acts to reduce surface cooling from the atmosphere, thereby the perception of warming - less cooling - of the layer below. This is a typical usage of thermal insulation. However, there is an analogy in the warming phase, in which snow cover acts to reduce warming of the layer below. For example, snow must be melt before the atmosphere directly warms the soil/ice/water. Of critical importance is that snow with high albedo increases an amount of the upward radiative flux at the surface prior to this direct

warming of the soil/ice/water below. In our model, as we discuss in Supplementary S3, the soil cooling is largely led by anomalous upward radiation flux at the surface level (i.e., a shortage of radiation flux into the soil) (Supplementary Fig. S3b), which is a result of anomalous additional snow cover (Fig. 5b).

We here present an additional analysis of sea ice and snow cover relationship using Northern Hemisphere EASE-Grid 2.0 Weekly Snow Cover and Sea Ice Extent, Version 4.0 (NSIDC-0046) for the period of 1979-2016. We obtained monthly snow cover converted from the weekly data. We calculated lag regressions between November sea ice anomalies over the Barents/Kara Seas and the snow cover extent in the following February-March-April. More snow anomalies are found in relation to the preceding sea ice loss over the mid-latitude regions except the western Russia and high elevation regions around Tibetan Plateau (Figure R2a). When we looked regressions using de-trended data, more snow anomalies are similarly found in the mid-latitudes (Figure R2b). These observation-based results indicate that anomalously large extent of snow cover in the marginal snow regions in the early spring emerges after BKS sea ice loss, and roughly support our simulation results.

In consistent with our understanding of the thermal insulation role of the snow cover, described above, we observe increased upward radiative flux (mostly from shortwave radiation, not shown) in early spring, which is followed by increased downward turbulent heat flux (see Supplementary S3).

We understand that our experiment is ideal such that it uses repeated annual forcing from the atmosphere and climatological SST. However, we would argue that information provided so far warrants a need for discussions on the role of snow cover and soil, which likely bring a noticeable memory to the Arctic climate system.

Figure R2. Observation-based relationship between sea ice variation and snow cover anomalies. Regression coefficients between November sea ice concentration over the Barents/Kara Seas (30–90°E, 65–85°N) and snow cover extent (SCE) in the following February-March-April from NSIDC-0046 for the period of 1979-2016. Regression coefficient is calculated **a**, without de-trend and **b**, with de-trend. Note that the sign of the coefficient is reversed to show anomalies associated with decrease of the sea ice. Hatch/double hatch indicate statistical significances exceeding 95/99%.

Perhaps I am misinterpreting the message from these results; if so, perhaps some clarification is needed in the text regarding these points. Given these potentially major concerns, I recommend that the manuscript undergo revision before acceptance. Additional analyses may be necessary, which could be included in supplemental material.

We have added the results of the additional analysis about observation-based soil temperature variation (Fig. R1 and the corresponding explanation) in Supplementary S4.

Specific comments

1. Line 121: add “(Methods)” after TEM diagnosis

We have made the suggested change (Line 132).

2. 147: Sentence refers to Eurasia, but Fig. 4b is for all longitudes, correct? If not, caption for Fig. 4b should be clarified.

Figure 4b is for all longitudes. We have modified the sentence accordingly (Line 472).

3. 153: This statement needs clarification. I suggest “Increased negative snowcover anomalies...”.

Here, we wanted to express that we found positive snow cover anomalies, corresponding to a greater than normal increase of snow cover over the mid-latitudes, whereas negative anomalies were found at the Arctic coastal latitudes. We have modified the sentence to more clearly express this intended meaning (Line 165).

4. 157: It looks like the anomaly weakens in autumn rather than summer – easier to see this in Fig. 3d.

The cold soil temperature anomaly weakens in both summer and autumn, and we have modified the relevant sentence to more clearly state this. It should be noted that the behavior of the soil temperature anomaly is slightly different than that of the near-surface temperature anomaly. We have therefore added Fig. 6, which shows the annual cycle of the soil temperature anomaly in the individual soil layers of the model. While near-surface soil layers vary closely with air temperature, the deeper layers lag by about a half year.

5. 160: insert “late” before spring?

We have added “late” as recommended (Line 173).

6. 161: How does this result relate to the fact that spring snowcover has been declining in recent decades?

As observed by satellite remote sensing, the snow cover over Eurasia has decreased in the past several decades. On the other hand, as we mentioned in the Discussion, some studies have suggested an increasing trend in the snow amount over Russia during the permanent snow cover season (e.g., Bulygina *et al.*, 2009).

Our experiment is highly idealized and is based on a repeated annual cycle that does not include the global warming effect. Therefore, the model results are not necessarily consistent with long-term trends in the real world. Observational data of increased snow amounts, even with ongoing global warming, could, however, be a supporting factor of our model result showing an increase of snow due to a decrease in sea ice.

We have modified a corresponding sentence to be more accurate (Line 240–245).

Reference

Bulygina, O. N., Razuvaev, V. N. & Korshunova, N. K. Changes in snow cover over Northern Eurasia in the last few decades. *Environ. Res. Lett.* **4**, 045026 (2009).

7. 167: Perhaps “disturbed” should be replaced by “removed as readily”?

We have modified the wording as you suggested (Line 200).

8. 176: Please add a reference describing the re-emergence of ocean temperature anomalies in winter.

We have added two citations, Alexander *et al.* (1999) and Taws *et al.* (2011), presenting the re-emergence of the winter SST anomaly in the North Pacific and North Atlantic, respectively (Line 209).

References

Alexander, M. A., Deser, C. & Timlin, M. S. The Reemergence of SST Anomalies in the North Pacific Ocean. *J. Clim.* **12**, 2419–2433 (1999).

Taws, S. L., Marsh, R., Wells, N. C. & Hirschi, J. Re-emerging ocean temperature anomalies in late-2010 associated with a repeat negative NAO. *Geophys. Res. Lett.* **38**, L20601 (2011).

9. 179: suggest removing “as a potential cause”

We have made the suggested change.

10. 181: suggest clarifying this statement by adding “turbulent-flux” before heating

We have made the clarification (Line 215).

11. 195: suggest removing “positive” (maybe negative ones, too) and make feedback plural – probably more than one feedback process.

We have made the suggested change (Line 226).

12. 196: make feedback plural

We have made the suggested change (Line 227).

13. 198-199: suggest rewording, such as: “...whether or not a repeated annual cycle is a realistic boundary condition – which might suppress or amplify the memory effect...”

This sentence might be unclear. We have modified sentence to “... *whether a repeated annual cycle or a historical variation is given as a boundary condition—which might suppress or amplify the memory effect from land processes.*” (Line 229–231).

14. 206: change “would” to “could”?

We have removed this sentence to avoid this supposition.

15. 209: Clarify “snow days” – does this mean the number of days in the year with snowcover or days when snow falls?

We wanted to express the duration of snow cover, but it was not an accurate expression.

We have modified the sentence to “*Consistent with our simulation, observations of Eurasian snow anomalies show an increase of snow amount during permanent snow cover seasons*” (Line 240–245).

16. 226: define AFES. What is temporal resolution of output from simulations?

We have revised the text to explain that AFES is an AGCM and that we used daily mean output (Line 262–265).

17. Fig. 1a: Are the initial atmospheric and land conditions for both serial experiments consistent with HICE? That is what the diagram implies. If so, perhaps the caption should include this detail.

Both the initial atmospheric and land conditions are identical for both serial experiments, but they are not consistent with HICE. This initial condition is from the climatology of the 1981–2010 period from the JRA25 reanalysis. Therefore, the original diagram we used in Fig. 1a was somewhat inaccurate. Although we think the initial condition of the serial experiments is not an important matter because we used the output after a 10-year spin up, we have modified Fig. 1 to remove this misleading diagram.

18. Fig. 1 and throughout text: The label “seasonal” for one of the initialized experiments seems somewhat confusing, as there are seasons in both sets of experiments. Perhaps consider calling it “sea ice” to distinguish it from the other experiments that isolate the land effects?

I suggest using shading in the row labeled as memory effect to help delineate these experiments. Also please label which sections of the schematic represent “total”. The caption for this figure could be a bit clearer, in my view. Following are some suggestions to help with that.

We have changed “seasonal” effect to “sea ice” effect throughout the manuscript. We have modified Fig. 1 to show the ensemble averaged state vectors and to clarify the compositions of sea ice and memory effects on total effect. We have also modified some of the other points as you suggested. We hope this change clarifies the explanation of our

unique experimental setting.

19. 393: before “HICE” add “atmospheric and land conditions existing at July 1st during”

We have made the suggested change (Line 448).

20. 395: add “of the atmospheric state” after “development”

We have modified the text as suggested (Line 450).

21. Fig. 3 caption: add (PCH) after polar cap height and (PCT) after polar cap temperature.

We have made the addition as suggested (Lines 470 and 471).

22. Fig. 4: Are these results only for Eurasia or for all longitudes? Please make a note in the caption, just to be clear.

They are all from all longitudes (an average). We have modified the caption to clarify this point (Line 478).

23. Fig. 5: I think it would be very helpful to add another set of plots for the “Total” response.

This would help understand how the seasonal and memory responses contribute to the total response. Hatch marks are very difficult to see – perhaps try white or yellow instead?

We have modified Fig. 5 to show the total effect.

Once again, we very much appreciate all of your constructive comments and suggestions. They helped us to improve the revised manuscript. We hope this revision addresses your concerns.

Reply to Reviewer 3:

Thank you very much for your comments. We have replied to your comments below and outlined the changes made to the revised manuscript. Please note that we changed from the use of the term “seasonal” effect, which referred to the half-year response to the Arctic sea ice loss, to the “sea ice” effect throughout the entire main text and supplementary files.

Reviewer #3 (Remarks to the Author):

The manuscript titled "Climate memory of Eurasian land processes for Arctic amplification" explores the role of Eurasian land processes in the Arctic warming and mid-latitude cooling by conducting a set of numerical experiments. Results from the serial experiments give full response of climate to the forcing of sea ice reduction, i.e. direct response to sea ice reduction plus feedbacks induced by the direct forcing. On the other hand, ensemble initialized experiments estimate the response of climate to sea ice reduction in the first year, which covers the fast component with a time scale less than a year and thus is assumed to be the direct response. The difference between the fast response and the full response is mainly due to the slow land processes and is attributed to the land memory effect by the authors. While this is a nice approach to separate the fast and slow components of the responses, a couple of issues need to be addressed.

1. The major concern from the reviewer is how does the memory component evolves on the inter-annual time scale. Since the authors define the memory component as the difference between the full response and the first year response, the we can only see the seasonal cycle of the memory component. However, it is also important to know when the memory component starts to develop and peak, its year-to-year evolution and its time scale character.

Thank you for this helpful comment. It was difficult to define the characteristics of the memory effect by the procedure described in our original manuscript. We thus performed additional experiments with “initialized runs” that extend the integration period by 3 years and have added this material to the Supplementary Materials. As we state in the main text, the circulation anomalies due to the sea ice effect bring cold anomalies over

Siberia during winter. This Siberian cooling forces cooling of the soil temperatures in the late winter every year (Fig. R1a). Cold soil temperature anomalies intensify year by year, and the coldness peak becomes later each year (see Fig. 6 in the main text). The largest difference between sea ice effects and the memory effect is reflected in the anomalous soil coldness in late spring to early autumn.

Next, we examined the causes of this anomalous soil coldness. With the sea ice effects, during late winter to early spring, anomalous soil coldness is largely led by the anomalous upward radiation forcing corresponding to a shortage of net short-wave radiation due to increased snow cover, that is, the albedo effect (the contribution of long-wave radiation is not shown separately, but it is small) (Fig. R1b, dashed lines). In turn, anomalous soil coldness cools the atmosphere through turbulent heat flux anomalies (Fig. R1b, solid lines).

In the memory effect, as a result of the delayed coldness peak, the anomalous soil coldness cools the atmosphere through turbulent heat flux during late spring. After that, heat flux anomalies are nearly zero during summer to autumn, while anomalous coldness gradually weakens. This indicates that the spring air temperature that has adjusted to the cold ground condition persists until autumn, and this condition possibly brings anomalous coldness and early snowfall in the autumn and early winter through the memory effect (Fig. 5 in the main text).

Such year-by-year accumulations and associated delays of the anomalous coldness, which is first activated by the circulation anomaly of sea ice effect, are important characteristics that form the memory effect. We have added a discussion of this analysis in Supplementary S3.

Figure R1. Development of *Sea ice* and *Memory* effects in the soil temperature. Seasonal evolutions of soil temperature anomalies and heat flux anomalies averaged over the Siberian/East Asian region ($90\text{--}120^\circ\text{E}$, $40\text{--}60^\circ\text{N}$). **a** Soil temperature of 0–4 m depth and **b** surface heat flux due to turbulent heat flux (solid lines) and surface radiation (sum of short-wave and long-wave radiation, dashed lines) are shown. For the heat flux anomalies, a positive anomaly indicates an increase of upward flux. Black, light blue, and blue indicate *Sea ice* effects from the first, second, and third year of the integration of the initialized experiment, respectively. Red indicates the *Memory* effect.

2. My second comment involves a couple of questions. Can we test the modeling results by comparing to observations? Can we detect the land memory component from observational data? Or, can we look at other model output (e.g. CMIP5) and see if they support the findings in this study or are just again what have been found here? This would make the results more relevant and robust.

We agree that a comparison with observed and/or CMIP model results would make our findings more relevant and robust. As an example, we conducted an additional analysis to examine observation evidence of the relationship between sea ice variation and ground temperature anomalies (see Supplementary S4). However, as we note in the manuscript, the observational evidence shows that the Eurasian continent has experienced serious warming in the past decade. This seems, at first glance, to be in opposition to our simulation results. On the other hand, such serious warming coincided with a moistening of soils in Eurasia, which might induce higher heat conductivity and make the memory effect more effective. There is, however, insufficient observational data on soil moisture that have continental-wide and multi-decade (at least two) coverage. Obtaining such data

will require international cooperation in the use/detection/observation of soil data.

Detecting and evaluating the memory effect in the other models (such as the CMIP models) would be quite useful to examine the model dependency and thus the relevance of our results, but this type of multi-model comparison is beyond our capacity or the focus of this study. We hope our findings will attract more interest in the land process in the context of the Arctic climate change and encourage international cooperation in future studies.

REVIEWERS' COMMENTS:

Reviewer #1 (Remarks to the Author):

Review: The role of Eurasian land processes as a climate memory in Arctic-mid-latitude climate linkage

Tetsu Nakamura, Koji Yamazaki, Tomonori Sato, Jinro Ukita

The authors have adequately responded to my questions and concerns and further improved their manuscript. Thus, I suggest to accept the article after making one minor change:

Line 13 and line 34 and elsewhere: Sorry to be petty here: I still think the term "mid-latitude cooling" is not correct, mid-latitudes as a whole are warming in winter and cooling occurs only somewhat more widespread over Siberia/ eastern Asia (see Koenigk and Fuentes Franco 2019, Int Journal of Climatology). I understand that you want to express that there is a connection between Barents Sea/ Kara Sea ice and cooling in parts of mid-latitudes in the winter thereafter (after detrending) but this is not getting clear from the way you write it.

I would suggest to call it Siberian or Central Asian cooling or as you do in line 111, mid-latitude anomaly (cooling) over the Siberian/ East Asian region.

Torben Koenigk

Reviewer #3 (Remarks to the Author):

The authors have addressed the concerns from the reviewer.

Reply to Reviewer 1 (Dr. Torben Koenigk):

Dear Dr. Koenigk,

We appreciate you for spending time for re-reviewing our manuscript. The manuscript was carefully revised by taking account of your comments.

Reviewer #1 (Remarks to the Author):

Review: The role of Eurasian land processes as a climate memory in Arctic-mid-latitude climate linkage

Tetsu Nakamura, Koji Yamazaki, Tomonori Sato, Jinro Ukita

The authors have adequately responded to my questions and concerns and further improved their manuscript. Thus, I suggest to accept the article after making one minor change:

Line 13 and line 34 and elsewhere: Sorry to be petty here: I still think the term “mid-latitude cooling” is not correct, mid-latitudes as a whole are warming in winter and cooling occurs only somewhat more widespread over Siberia/ eastern Asia (see Koenigk and Fuentes Franco 2019, Int Journal of Climatology). I understand that you want to express that there is a connection between Barents Sea/ Kara Sea ice and cooling in parts of mid-latitudes in the winter thereafter (after detrending) but this is not getting clear from the way you write it.

I would suggest to call it Siberian or Central Asian cooling or as you do in line 111, mid-latitude anomaly (cooling) over the Siberian/ East Asian region.

Torben Koenigk

We modified corresponding sentences to specify the location of cooling anomalies appropriately.

Line14: cooling over the Siberian/East Asian regions

Line 32-33: wintertime mid-latitude cooling over the Siberian/East Asian regions

Line 85: Eurasian mid-latitude cooling anomalies

Line 155-156: persistent cold anomalies over the Eurasian continent